# Localized Corrosion of Stainless Steel Triggered by Typical Inclusions in NaCl Solution: Oxy-Sulfide and MnS

**DOI:** 10.3390/ma16124323

**Published:** 2023-06-12

**Authors:** Dan Li, Hui Zhi, Zhaogui Lai, Ying Jin

**Affiliations:** National Center for Materials Service Safety, University of Science and Technology Beijing, Beijing 100083, China; lidan9203@126.com (D.L.); neu3dmaker@163.com (H.Z.); laizhaogui@ustb.edu.cn (Z.L.)

**Keywords:** oxy-sulfide inclusion, MnS, micro electrochemical test, AFM

## Abstract

The localized corrosion behavior of stainless steel (SS) induced by typical inclusion such as MnS and oxy-sulfide in NaCl solution was investigated by immersion tests and microelectrochemical tests. Oxy-sulfide consists of an internal polygonal oxide part and an external sulfide part. The surface Volta potential of the sulfide part is always lower than that of the surrounding matrix, as in the case of individual MnS, while the potential of the oxide part is indistinguishable from that of the surrounding matrix. Sulfides are soluble, while oxides are almost insoluble. Oxy-sulfide exhibits a complex electrochemical behavior in the passive region, which can be attributed to its complex composition and multi-interface coupling effects. It was found that both MnS and oxy-sulfide increase the susceptibility of the local area to pitting corrosion.

## 1. Introduction

It is well known that passive metals are usually prone to localized corrosion due to some local heterogeneities, such as non-metallic inclusions, which can disrupt the integrity of the passive film formed on the metal surface [1,2,3]. For most stainless steel (SS), the common inclusions can be classified into two types: sulfide (MnS) and oxide [4,5,6]. MnS is generally considered as a nucleation site for SS pitting corrosion in chlorine environments, and its corrosion and subsequently induced matrix corrosion have been extensively studied by many researchers [7,8,9]. Accordingly, a comprehensive mechanism for the MnS-induced pitting corrosion in chloride-containing solutions has been proposed in detail. First, under the attack of chloride ions, MnS starts to dissolve from its boundary region and release sulfur. Then, the adjacent active matrix is exposed due to the dissolution of MnS, forming trenches along the MnS/matrix boundary. The synergistic effect of sulfur and chloride ions deteriorates the trench environment and promotes the dissolution of MnS and the peripheral active matrix, which may lead to metastable pitting or even stable pitting [4,10,11,12].

Oxides have not received much attention in terms of corrosion behavior compared to sulfides because they have a more pronounced effect on the mechanical properties of steels, and on the other hand, they are always considered to be inert. The oxide composition is complex and varies with heat treatment temperature [13,14,15]. A brief summary of the oxides formed in stainless steel reported in the literature is given in Table 1. The underlying reason for the variation in composition is the reaction of O with other alloying elements in stainless steel, such as Cr, Mn, and Al [13]. During the heat treatment of 304 stainless steel, MnO-SiO_2_ inclusions may be transformed into MnO-Cr_2_O_3_ inclusions, because Cr can react with manganese silicate to form MnO-Cr_2_O_3_, and the higher the heat treatment temperature (1000 °C to 1200 °C), the higher transformation rate of manganese silicate to manganese chromite [15]. Furthermore, the heat treatment time also has an effect on the composition of the oxide inclusions. Y. Ren [6] stated that the composite inclusion MnO-SiO_2_(-CrO_x_-Al_2_O_3_) transformed to MnO-SiO_2_(-CrO_x_-MgO-Al_2_O_3_) after 15 min of heat treatment and then evolved to MnO-Cr_2_O_3_ after 60 min. In addition to the effect of heat treatment, the oxide composition is also influenced by inclusion size. They also concluded that smaller inclusions tend to transform completely, while larger inclusions are incompletely converted, leaving a trace of Si. Although oxide formation is affected by various factors, no matter what oxides are formed in SS, they are extremely detrimental to mechanical properties such as strength, toughness, and fatigue resistance [16]. Al_2_O_3_-rich or Al_2_O_3_-MgO-rich inclusions have been reported to cause liner defects on SS surfaces due to their high hardness [17]. In fact, the effect of these oxides on the corrosion performance of SS should not be neglected, as they may also cause the pitting of SS.

It was earlier suggested that pits always nucleate at sulfide inclusions rather than oxide inclusions [18]. However, a different view was held by Wranglen [4], who stated that both sulfide and oxide may induce pitting corrosion. M. A. Baker [19] found that dissolution started from the interface of mixed oxide inclusion and matrix in stainless after being exposed to H_2_SO_4_ + NaCl + H_2_O_2_ solution for 10 s. Heon Young Ha [20] reported that non-metallic oxide inclusions in ferritic stainless steel can initiate pitting corrosion. S. Zheng [5] also reported that (Mg, Al, Ca)-oxide inclusions can lead to pitting corrosion of 316L stainless steel exposed to sulfur environments containing chloride ions. On the other hand, and more importantly, oxides tend to combine with sulfides susceptible to chloride ions to form kinds of mixed inclusions in stainless steel [21]. This can greatly increase the probability of oxide inclusion nucleation for pitting corrosion. J. E. Castle [22] observed that corrosion attack starts with MnS inclusions and MnS precipitates associated with oxide inclusions. G. Bai et al. found that pitting is always initiated at the interfaces between austenite matrix and oxide/sulfide inclusions [23]. S. J. Zheng et al. [24] even explored the relationship between oxide and MnS and pointed out that the coupling of oxide particles and MnS generates local nano-galvanic cells, leading to the preferential dissolution of MnS. It can be seen that the possible pitting nucleation sites in stainless steel include not only MnS but also oxides or mixed oxide/sulfide inclusions. Therefore, it is necessary to reconceptualize the problem to determine whether these two inclusions are definitely nucleation sites and to assess the susceptibility of both inclusions to localized corrosion.

Localized corrosion triggered by inclusions can be well characterized by microelectrochemical techniques, while changes in current density during electrochemical testing can be used to measure the susceptibility of localized areas to corrosion [25,26,27]. Böhni et al. [28] pointed out that the current transients due to localized activation and repassivation processes in the passive potential range can be easily measured through a microelectrochemical cell. Webb et al. [26] investigated the dissolution behavior of individual sulfide inclusion within a microcell and concluded that the increase in current is largely due to MnS dissolution, while the decrease in current is due to repassivation around MnS. However, fewer similar studies have been performed for oxides, and it is not clear whether oxides exhibit current change phenomena in the passive region similar to that of MnS.

Therefore, this paper presents a comparative study of the local corrosion behavior of two typical inclusions initiated in stainless steel through immersion tests and attempts to elucidate the role of MnS and oxy-sulfide in initiating pitting corrosion and their effect on the susceptibility of the matrix to pitting corrosion from a microscopic perspective. The results suggest that both inclusions may be pitting nucleation sites and can significantly increase the susceptibility of local areas to pitting. Specifically, immersion tests show that both MnS and sulfide parts of an oxy-sulfide are soluble in NaCl solution, while the oxides are almost insoluble. The oxide part of an oxy-sulfide may dissolve slightly only when the combined sulfide part causes pitting corrosion. In contrast to MnS, oxy-sulfide exhibits complex electrochemical behavior in the passive region due to its complex components and multi-interface structure.

**Table 1 materials-16-04323-t001:** Brief summary of oxide inclusion type in stainless steel reported in the literature.

Steel Type	Inclusion Type (Composition)	Ref.
316L stainless steel	(Mg, Al, Ca) oxides	[5]
304 stainless steel	MnO·SiO_2_(CrO_x_-Al_2_O_3_)	[6]
Austenitic stainless steel	CaO·SiO_2_·Al_2_O_3_·MnO·MgO	[13]
18Cr-8Ni stainless steel	47MnO-47SiO_2_-6Cr_2_O_3_	[15]
Si-Mn killed 304 stainless steel	Al_2_O_3_-SiO_2_-MnO-CaO	[17]
Stainless steel	AI/Ti/Mn/Cr oxide inclusions	[19]
Saw wire steels	dual-phase (MnO-SiO_2_-Al_2_O_3_) + (SiO_2_) inclusions	[29]
Si-killed stainless steel	Al_2_O_3_-SiO_2_-CaO-MnO	[30]
Ferrite stainless steel	MgO·Al_2_O_3_-CaO-TiO_x_	[31]
316L stainless steel	SiO_2_-Cr_2_O_3_-MnO	[32]
Y addition stainless steel	(SiO_2_-Y_2_O_3_)–(MnO-Cr_2_O_3_-Al_2_O_3_)	[16]
Si-deoxidized 18Cr-8Ni stainless steels	Al_2_O_3_-MgO-SiO_2_-MnO	[33]
304 stainless steel	(CaO-SiO_2_-Al_2_O_3_-MgO)–MgO·Al_2_O_3_	[34]
Al–Ti–Ca complex deoxidized steel	Al–Ti–Ca–O inclusion	[35]
SUH 409L stainless steel	MgO·Al_2_O_3_	[36]

## 2. Experimental and Materials

### 2.1. Sample Preparation

A re-sulfurized (sulfur addition) type 304 stainless steel of the composition given in Table 2 was used as a test material, and the sample was forged and rolled prior to heat treatment at 1080 °C. Samples were prepared as 10 × 10 × 3 mm specimens (the cutting direction parallel to the rolling direction), ground using 3000 grit emery paper, and polished with 1μm diamond paste before experimentation. The following experiments were performed at room temperature (25 ± 1 °C) with a relative humidity of about 50%.

### 2.2. Surface Analyses

Inclusion morphology and elemental information were obtained by field emission scanning electron microscope (FE-SEM, Merlin Compact, Zeiss, Jena, Germany) and a coupled energy dispersive spectrometer (EDS, Oxford X-max, Abingdon, UK), respectively. Both the second electron (SE) and back-scattered electron (BSE) modes were used with an acceleration voltage of 15 kV and a working distance of about 11 mm.

Three-dimensional topography and line scan profiles of the dissolved inclusions were mapped with an atomic force microscope (AFM, Dimension Icon, Bruker, Karlsruhe, Germany). The surface Volta potential of the microregion-containing inclusion was tested by scanning Kelvin probe force microscopy (SKPFM, one mode of AFM, Nanosurf AG, Liestal, Switzerland) using a conductive probe (SCM-PIT). AFM topography and corresponding surface Volta potential were imaged at a scan rate of about 1 Hz and a line scan resolution of 512 × 512. The scan size of AFM maps depends on the size of inclusion in the test areas, usually a few microns square.

In addition to EDS, the composition of local areas containing single inclusions can be detected by a laser confocal Raman microscope (LCRM, Alpha 300+, WITec, Ulm, Germany). During the detection, a laser with a laser wavelength of 532 nm was used with a laser power of approximately 1 mW and a grating of 600 g/mm, and the data collection time was about 6 min.

### 2.3. Corrosion Tests

#### 2.3.1. Immersion Test

Two immersion periods of 60 h and 7 days were designed and performed in order to obtain the morphological changes of the inclusions. Prior to the experiments, the approximate locations of the inclusions in the specimens were recorded with an optical microscope, and diamond notches were made around these inclusions with a microhardness tester to locate the inclusions. The test solution in this work was a 3.5 wt.% NaCl solution prepared from analytical-grade sodium chloride and deionized water.

#### 2.3.2. Microelectrochemical Test

The microelectrochemical tests were conducted using a homemade electrochemical microcell system (schematic diagram of the device is shown in Appendix A) [37,38], an effective device for exploring the corrosion behavior of local areas containing different types of inclusions. The test system consists of an electrochemical workstation (Gamry 600+), an optical microscope (Olympus BX53, Shinjuku, Japan) equipped with an automatic platform and a computer. In the experiments, only a small part of the sample surface was exposed to the electrolyte, and the rest was coated with a photoresist with a thickness of about 2.5 μm, see Appendix A (example image). A three-electrode system was employed, containing an Ag/AgCl (saturated KCl) reference electrode (RE), counter electrode (CE, platinum wire), and the working electrode (WE, a circular area of 20 μm diameter). Potentiodynamic polarization tests were conducted in the potential range of −400~800 mV (vs. Ag/AgCl) with a scan rate of 1 mV/s.

## 3. Results and Discussion

### 3.1. Inclusion Characterizations

#### 3.1.1. FE-SEM Analyses

The OM and SEM morphological characteristics of the inclusions formed in the specimens are shown in Figure 1, with two main shapes of inclusions: rod-shaped and subspherical. In terms of shape and color, the rod-shaped inclusions (Figure 1c) consist of a single component, while the subspherical inclusions (Figure 1d) are composed of two parts: an external rounded part and an internal polygonal part. The results of the EDS mapping in Figure 2 indicate that the rod inclusions are manganese sulfide (MnS, with an atomic ratio of Mn to S of about 1:1). For the subspherical inclusion, the EDS mapping (Figure 3) shows that it contains multiple elements, with Cr, Al, Mn, Ti, and O forming a polygonal oxide part and S, Mn forming a rounded sulfide part. During solidification, aluminum has a strong affinity for oxygen in liquid iron, forming oxide particles that are then inclined to agglomerate together. On the other hand, Al-containing oxides also combine with S and Mn to form mixed inclusions, and these oxides are always the nucleation core of the MnS in molten steel during inclusion, forming mixed TiO_2_-MnO-Al_2_O_3_-Cr_2_O_3_ inclusions [21].

Prior to inclusion characterization, statistical experiments were performed on several specimens to obtain general information on the distribution and composition of the non-metallic inclusions formed in re-sulfurized stainless steel (see Appendix A). The results showed that, in terms of the observed area, the MnS inclusions are all rod-shaped, with most of them ranging from around 2 to 13 μm in length, and the polygonal oxides are bound to sulfides in the form of being wrapped by spherical sulfides or directly embedded in rod-shaped sulfides, forming mixed inclusions of oxy-sulfides, as shown in Appendix A. As we mentioned in the experimental part, the specimens were subjected to rolling treatment, resulting in the extension of the inclusions in the rolling direction and the formation of rod-shaped MnS. The polygonal oxides (Cr/Al/Mn/Ti/O), however, maintain the polygonal geometry due to their high hardness and lack of deformation. Since oxy-sulfide is a mixture containing multiple elements and its chemical structure is difficult to determine, it will be considered as a whole in this paper to assess its effect on localized corrosion. Thus, the inclusions in the specimens can be divided into two types depending on the chemical composition: MnS and oxy-sulfide, which can be clearly identified by different apparatuses depending on the geometry.

#### 3.1.2. SKPFM Measurements

Figure 4 shows the topographic maps of the two inclusions and the corresponding surface Volta potential distribution. MnS inclusions (Figure 4a) have lower surface Volta potential than the surrounding matrix, and the potential difference between them is about 123 mV (Figure 4c). For the oxy-sulfide inclusion (Figure 4b), the surface Volta potential of the sulfide part is lower than that of the surrounding matrix, while the surface potential of the oxide is close to that of the surrounding matrix (Figure 4d).

It is widely accepted that the potential difference between the two phases is the driving force for microgalvanic corrosion. The potential difference between the MnS/sulfides and the matrix suggests that microgalvanic corrosion tends to occur around it, where the MnS/sulfides with lower surface Volta potential will act as anodes and preferentially dissolve. From another perspective, this also implies that the passive film on the SS surface is heterogeneous, and it is weak or even discontinuous at MnS/sulfides, thus exhibiting different surface Volta potentials. Therefore, in a corrosive environment, such as a chloride-containing solution, the anion will preferentially adsorb on MnS/sulfides lacking passive film protection and promoting its anodic dissolution. This is in agreement with many studies on MnS-induced pitting corrosion, which tends to occur as a nucleation point for pitting corrosion [1,2]. In the case of oxides, they exhibit the same potential distribution as the surrounding matrix, which means that the oxides themselves do not actively induce microgalvanic corrosion. M.A. Baker et al. [19] suggested that oxides in stainless steel are electrically inert. Therefore, it can be inferred that in oxy-sulfide inclusions, the sulfide part acts as an anode and preferentially dissolves in a similar manner as a single MnS inclusion, while the oxide part may act as a cathode in a similar manner to a matrix with passive film or may only exist as an inert part without any effect.

### 3.2. Immersion Tests

In order to gain a preliminary understanding of the dissolution behavior of these two inclusions, immersion tests were performed. Figure 5 provides the SEM morphology of a set of marked inclusions before and after 60 h of immersion in NaCl solution. It can be seen that both the MnS inclusions and the sulfide parts of oxy-sulfide inclusions were dissolved, while the oxide parts kept their original shape almost without dissolution. A more interesting point is that the corrosion attack always starts from the edge areas of individual MnS inclusions, whereas for oxy-sulfide inclusions, the corrosion attack, although starting from the sulfide edge, tends to start from the edge on the oxide side rather than from the side bordering the matrix (See Appendix A). This dissolution behavior of individual MnS is consistent with our previous studies on MnS-induced localized corrosion, where MnS as a pitting nucleation site always starts dissolving in its boundary area [39]. The preferential dissolution of individual MnS from its edge area is driven by the microgalvanic effect between MnS and the surrounding matrix. Therefore, it is assumed that in the oxy-sulfide, the preferential dissolution of the sulfide boundary area near the oxide side may be driven by the microgalvanic effect between the sulfide part and the oxide part.

Subsequently, another group of marked inclusions was subjected to relatively long-term immersion tests for about 7 days, and the results are shown in Figure 6. The marked MnS (Figure 6a) was completely dissolved (Figure 6b), forming a corrosion pit with a depth of approximately 400 nm (Figure 6c–e). According to the size of the pit mouth, the size of the pit was the same as that of the original MnS, indicating that the dissolution of MnS neither triggered the dissolution of the surrounding matrix nor destroyed the passive film on the surrounding matrix. In other words, the dissolution of MnS did not induce pitting. In contrast, the marked oxy-sulfide inclusion (Figure 6f) dissolved in the sulfide part, while the oxide dissolved only slightly (the area enclosed by the green dotted line in Figure 6g), forming a corrosion pit with a depth of about 170 nm. Notably, the oxide falls inside the corrosion pit leading to the exposure of the other side of the oxide (Figure 6h–j), indicating that the dissolution of sulfide beneath the oxide caused the collapse below the oxide. Moreover, the pit profile is larger than the original inclusion, which means that the matrix around the inclusion was also dissolved, implying that the oxy-sulfide caused the pitting corrosion. The dissolution phenomenon of oxy-sulfide indicates that oxide is a polyhedron, soluble, and may dissolve when corrosion around it is severe.

The above immersion tests indicate that either individual MnS inclusions or the sulfide parts of oxy-sulfide inclusions dissolve in NaCl solution. This is consistent with SKPFM’s inference that MnS/sulfides susceptible to chloride ions are readily dissolved, while oxides exhibit resistance to chloride ion attack. The dissolution phenomenon suggests that not all MnS induces pitting corrosion; if the local environment formed by MnS dissolution is not sufficient to sustain the dissolution of the active matrix, the dissolution process is suppressed and passivation occurs. In contrast, as long as the local environment formed by MnS dissolution can inhibit matrix passivation and the matrix continues to dissolve, oxy-sulfide can also induce pitting corrosion. It is generally believed that sulfur species from MnS dissolution and chloride ions concentrate in the trenches formed at the MnS/matrix boundary, lowering the local pH and leading to an aggressive solution component in the local environment where pitting corrosion nucleates and propagates [40,41,42]. The problem is that there is a critical value for the concentration of aggressive solution component, below which, passivation occurs [11,43,44]. That is, the occurrence of pitting corrosion depends on the degree of aggressiveness of the local environment created by MnS dissolution. Both MnS and oxy-sulfide can induce pitting corrosion as long as the local environment created by MnS/sulfide dissolution is sufficiently hostile, and the oxide may dissolve once a corrosive environment is formed.

In addition, Raman detection was performed on the inclusion area before and after immersion, and the results are shown in Figure 7. As can be seen from Figure 7a that there is only one strong peak located at 279 cm^−1^ before immersion, which is the characteristic Raman line of MnS [45]. After immersion for about 7 d, the MnS peak disappeared and a new peak appeared at 471 cm^−1^, which belongs to elemental sulfur [46]. The disappearance of the MnS peak implies the dissolution of MnS, leaving elemental sulfur as a product in the corrosion pit, which is consistent with the literature [7,10]. For oxy-sulfide inclusion, except for the MnS peak at 279 cm^−1^, there is a set of peaks in the range of 500–700 cm^−1^, representing oxides. Unfortunately, it is hard to separate these peaks for analysis and they are not quite the same as the Raman peaks reported in the literature for Cr oxide, Al oxide, Ti oxide, Mn oxide, etc. This is supposed to be due to the complex composition and structure of the oxide. After immersion, the MnS peak disappeared and the oxide peaks were almost unchanged, indicating that the sulfide was dissolved and the oxide was still present. The Raman results are consistent with the SEM/AFM results, confirming the dissolution of MnS and the sulfide part.

### 3.3. Microelectrochemical Tests

Microelectrochemical tests were conducted on stainless steel matrix micro-regions containing single inclusions. The tests included a matrix containing a single MnS inclusion (Figure 8b), a matrix containing a single oxy-sulfide inclusion (Figure 8c), and a blank region of the matrix only as a comparison (Figure 8a). Figure 8d shows a typical polarization curve for an SS matrix with a smooth passive region, indicating that the electrode surface is stable, i.e., the passive film on the matrix is intact. The trend of the polarization curve (Figure 8e) for the matrix containing MnS is closer to that of the matrix only, except that there is a current transient in the passive region. The sudden increase in current density is due to MnS dissolution, while the decrease in current density is due to the repassivation of the matrix. According to other researchers, MnS dissolution may induce metastable pitting, leading to an increase in the current density and thus forming a bulge on the passive region. Here, the MnS dissolution only caused a transient in the current density, indicating that MnS only dissolved itself without causing metastable pitting. It was reported that MnS is thermodynamically unstable in the passive region [7], which is verified by the change in the current density in the passive region of the polarization curve, indicating that MnS can dissolve at low anodic potentials of about −150 mV.

In contrast to the polarization curves of the first two, the polarization curve of the matrix containing single oxy-sulfide (Figure 8f) exhibits significant current fluctuations in the passive region. In this case, the increase in current density occurs first at a potential of about −200 mV, then immediately drops even below the passivation current density, and finally returns to the passive region. With increasing potential, the changes in current density become frequent and small in magnitude. The increase in current density is due to the dissolution of inclusion, while the subsequent decrease in the current density is due to the immediate repassivation of the matrix. Among these, the inclusion dissolution contains not only sulfide dissolution but also oxide dissolution. The multiple fluctuations of the current in the passive region suggest the alternation of inclusion dissolution and matrix repassivation processes, which can be attributed to the complex structure of the oxy-sulfide inclusion with multiple interfaces, leading to complicated microgalvanic effects.

Indeed, the different electrochemical behavior between the two inclusions as well as the matrix can be attributed to various factors such as chemical composition, structure, and surface state (passive film integrity). The electrochemical behavior of mixed inclusions in the passive region is complex and includes oxide dissolution as well as matrix repassivation in addition to sulfide dissolution. It has been reported that the oxide dissolution depends on its chemical property, i.e., solubility, and a decrease in pH in the local region can lead to a partial dissolution of generally inert oxide inclusions, and Al^3+^, Cr^3+^, and Mn^2+^ might be released through oxide dissolution when pH < 3 [19]. As mentioned earlier, the anodic dissolution of MnS/sulfides creates a local acidic environment with low pH, which may then result in oxide dissolution, as manifested by multiple increases in the current density in the passive region. Afterwards, the dissolved oxides may, in turn, facilitate the repassivation process as their components, such as elemental Cr and Ti, favor the corrosion resistance of the local region and promote the repair of the passive film, as shown by an immediate decrease in current density after a small increase [35]. Furthermore, the preferential dissolution of the sulfide parts may form corrosion pits, leaving the oxide parts in the pits and exposing most of the oxide (see Figure 6d), resulting in an increase in the electrode surface and consequently a decrease in current density to a level lower than passivation current density.

A comparative result of these three polarization curves is shown in Figure 9, which shows approximately the same trend except for the current fluctuations in the passive region, and their current densities start to increase at about 0.3 V. At this time, the increase in current density implies that passive film on the local area is broken down. The reason for the same trend in the three curves is that most of the tested area is the matrix, so the obtained electrode surface information is largely influenced by the matrix. It can be seen from Table 3 that both *E_corr_* and *i_corr_* increase in the following order: *E_MnS_* < *E_matrix_* < *E_oxy-sulfide_*, *i_oxy-sulfide_* < *i_matrix_* < *i_MnS_*. In general, the lowest corrosion potential and highest corrosion current density indicate the lowest local corrosion resistance of the electrode surface, i.e., the lowest local corrosion resistance of MnS. It should be noted here that the fitted value of the three sets of data have little difference and fitting errors are inevitable, so the *E_corr_* and *i_corr_* here can only be used as a reference. The microelectrochemical behavior for both inclusions needs to be further investigated to make precise conclusions. In future work, not only should the complete localized corrosion mechanism in stainless steel be refined by more precise instruments or simulation/calculations, such as high-resolution transmission electron microscopy and DFT calculations [47], but corresponding corrosion protection methods should also be explored, such as the application of corrosion inhibitors on stainless steel [48,49].

Moreover, it is worth mentioning that the current density of the matrix containing both inclusions increase faster than that of the matrix only after 0.3 V (area enclosed by yellow dotted lines in Figure 9), which may be due to the propagation of metastable pitting induced by inclusions. As known, a complete pitting process includes nucleation, metastable pitting propagation, and stable pitting growth [11]. Pitting is always considered to be nucleated at inclusive particles in stainless steel. Metastable pitting occurs when the current density continues to increase but before the pitting potential is reached (the pitting potential corresponds to a current density value of about 10^−4^ A·cm^−2^). With continuing increases in anodic potential, metastable pitting may evolve into stable pitting. This phenomenon suggests that the presence of both MnS and oxy-sulfide inclusion increases the susceptibility of local areas to pitting corrosion.

Based on the above experimental results, a brief dissolution schematic diagram of MnS and oxy−sulfide inclusion is drawn in Figure 10. It is found that whether pitting corrosion is triggered by inclusions depends on the local environment created by sulfide dissolution. If the local acidic environment is not harsh enough to sustain the dissolution of the active matrix, passivation occurs, and the MnS only dissolves itself; conversely, the local environment is harsh enough that the matrix passivation is suppressed and the dissolution of sulfide not only induces pitting corrosion but may also cause the slight dissolution of the oxide. The microelectrochemical results indicate that both MnS and oxy-sulfide inclusions have a significant effect on the SS matrix and can increase the susceptibility of the matrix to pitting corrosion.

## 4. Summary

In this work, the localized corrosion behaviors of stainless steel in NaCl solution caused by typical inclusions were investigated. The following conclusions can be drawn:(1)Two types of inclusions are formed in the stainless steel: MnS and oxy-sulfide inclusion, where oxy-sulfide consists of a polygonal oxide part (Al/Cr/Mn/Ti/O) and a round sulfide (MnS) part.(2)In oxy-sulfide, the surface Volta potential of sulfide is lower than that of the surrounding matrix, while the surface Volta potential of oxide does not differ much from that of the surrounding matrix.(3)Both the MnS and sulfide parts of oxy-sulfide are easily dissolved in NaCl solution, while the oxide part of oxy-sulfide is almost insoluble.(4)The matrix containing oxy-sulfide inclusion has multiple current fluctuations in the passive region compared to the matrix with only the matrix and the matrix containing MnS, which may be attributed to the multi-interface coupling effect caused by the complex composition of the oxide. This also indicates that the dissolution of oxy-sulfide inclusion is complex, including not only the sulfide dissolution and matrix repassivation but also the oxide dissolution.

## Figures and Tables

**Figure 1 materials-16-04323-f001:**
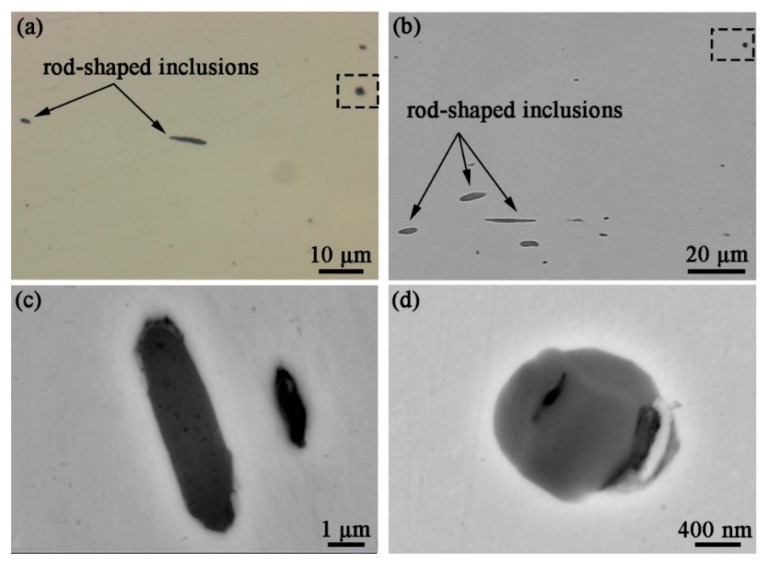
(**a**) OM and (**b**) SEM morphology of the inclusions formed in re-sulfurized stainless steel; SEM morphology of (**c**) MnS and (**d**) oxy-sulfide.

**Figure 2 materials-16-04323-f002:**
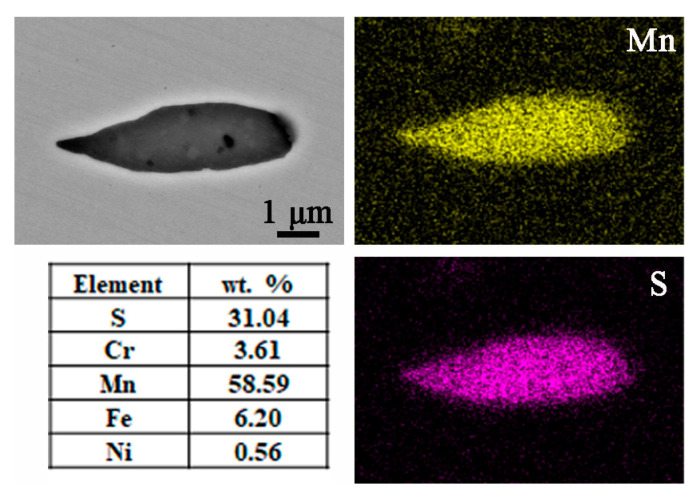
SEM morphology and EDS mapping of the rod-shaped MnS inclusion.

**Figure 3 materials-16-04323-f003:**
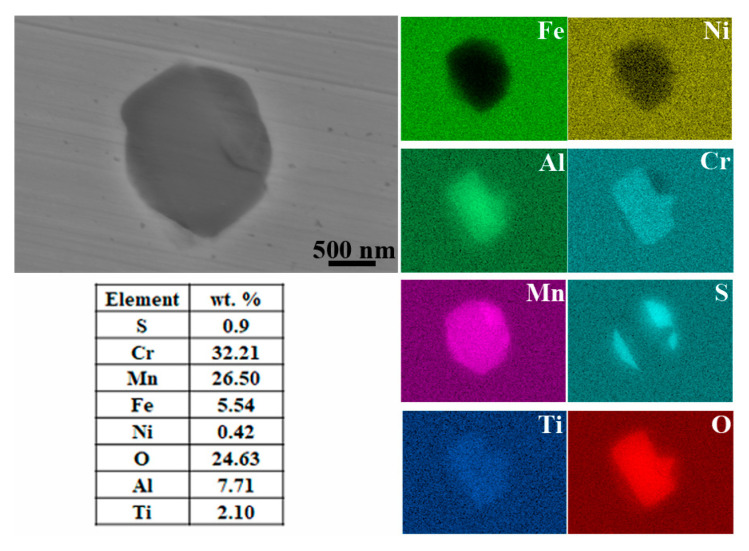
SEM morphology and EDS mapping of the oxy-sulfide inclusion.

**Figure 4 materials-16-04323-f004:**
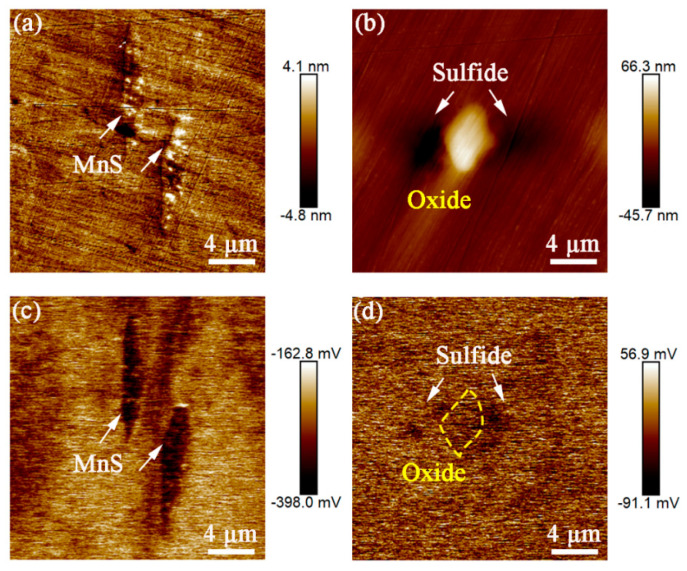
AFM topography of (**a**) MnS and (**b**) oxy−sulfide inclusion and the corresponding surface Volta potential maps of (**c**) MnS and (**d**) oxy-sulfide.

**Figure 5 materials-16-04323-f005:**
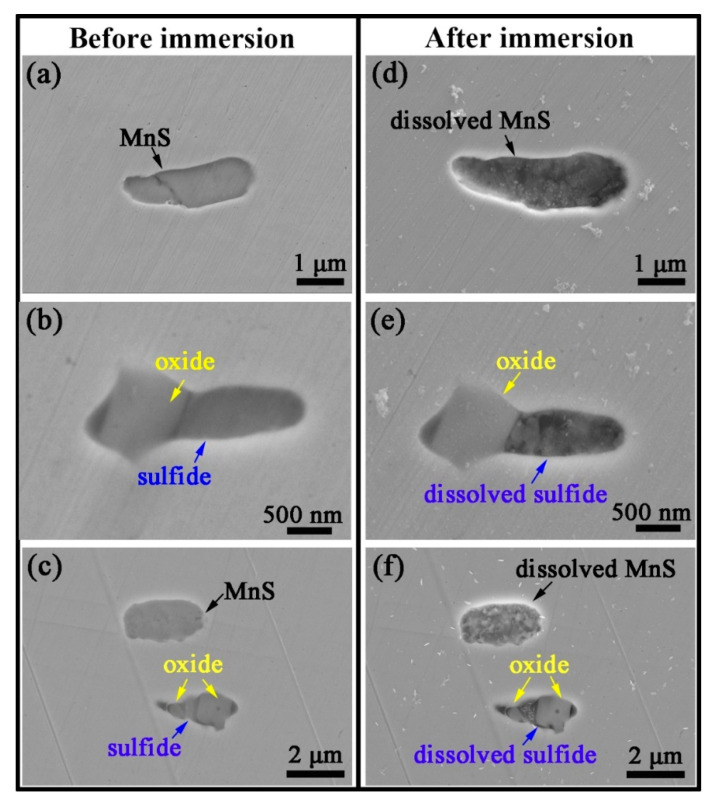
SEM morphologies of the inclusions (**a**–**c**) before and (**d**–**f**) after immersion for 60 h.

**Figure 6 materials-16-04323-f006:**
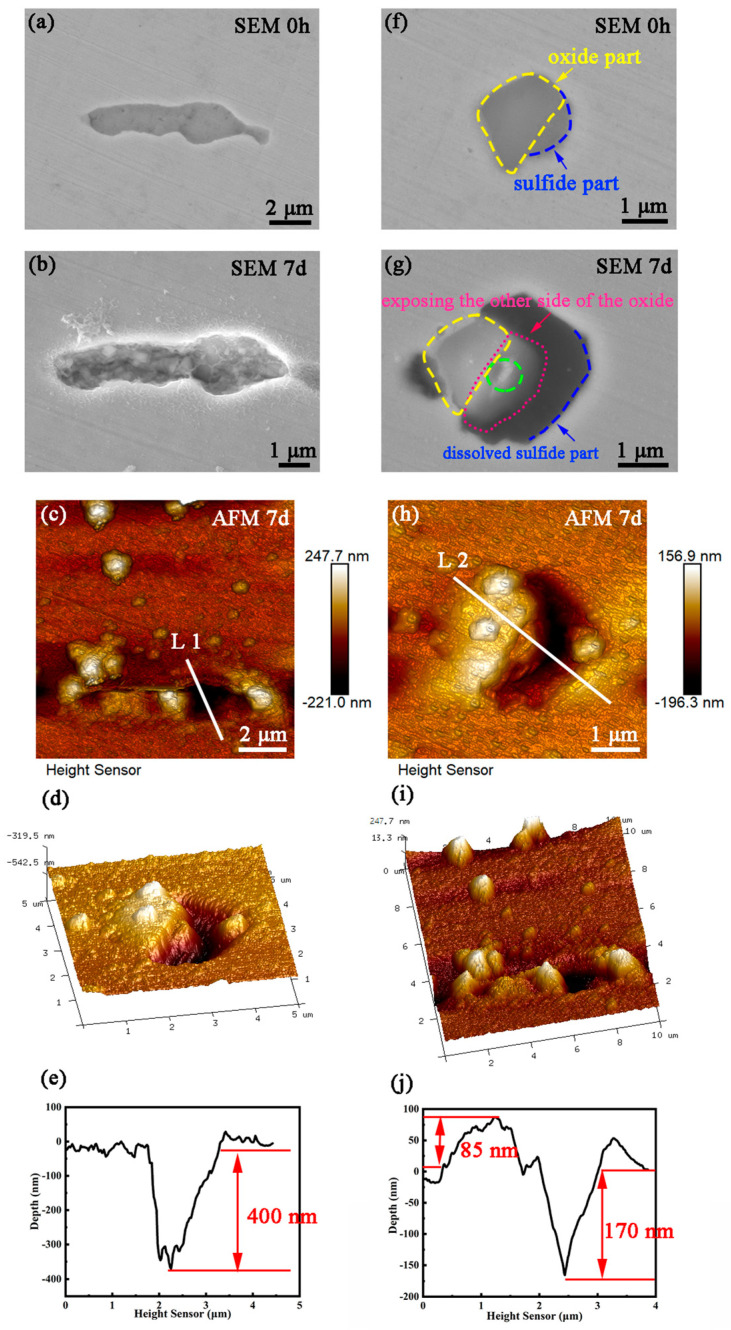
(**a**) SEM morphology of MnS inclusion before immersion, (**b**) SEM morphology, (**c**–**e**) the corresponding AFM topography and height profiles of the MnS after immersion for 7 d; (**f**) SEM morphology of oxy-sulfide inclusion before immersion, (**g**) SEM morphology, (**h**–**j**) the corresponding AFM topography and height profiles of the oxy-sulfide inclusion after immersion for 7 d.

**Figure 7 materials-16-04323-f007:**
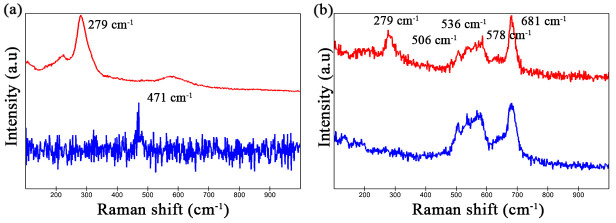
Raman lines of the marked MnS shown in Figure 6a and oxy−sulfide inclusion shown in Figure 6f before and after immersion for 7 d. (**a**) Only one peak of 279 cm^−1^ at 0 h is characteristic Raman peak of MnS, and the weak peak of 471 cm^−1^ appearing after immersion for 7 d represents elemental sulfur. (**b**) The group of Raman peaks located at 500–700 cm^−1^ belongs to oxides.

**Figure 8 materials-16-04323-f008:**
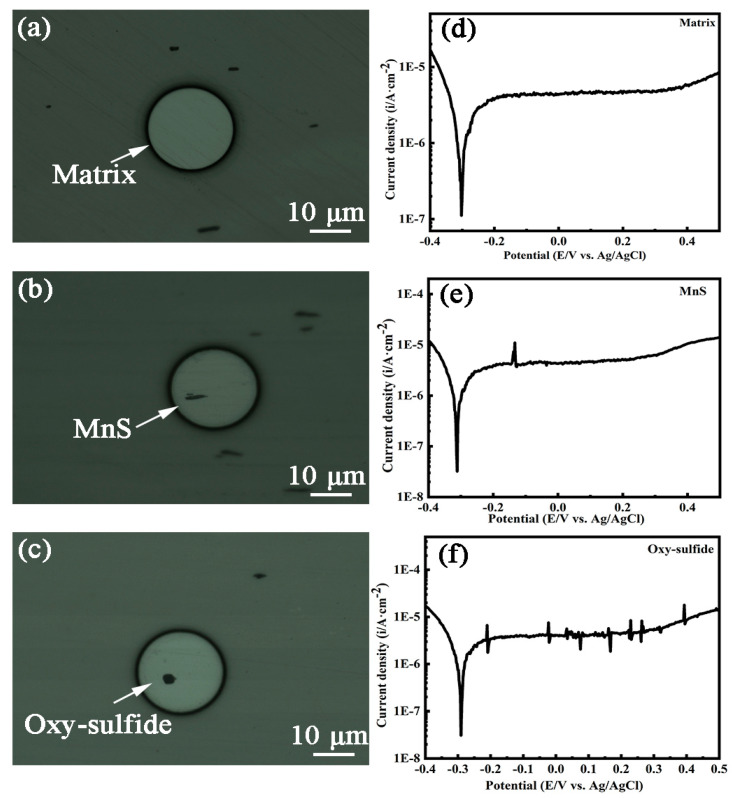
OM morphology of the tested micro areas and the corresponding polarization curves for each tested area. (**a**,**d**) Matrix only; (**b**,**e**) matrix containing MnS inclusion; (**c**,**f**) matrix containing oxy−sulfide inclusion.

**Figure 9 materials-16-04323-f009:**
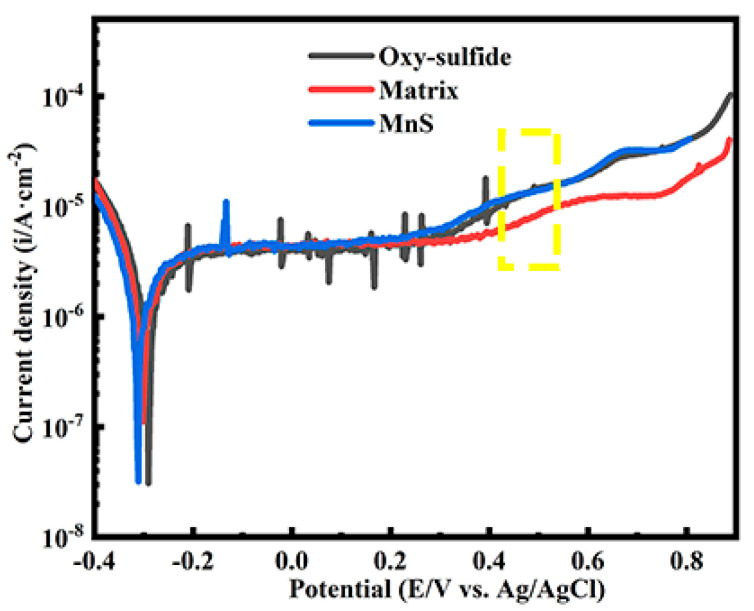
Comparative result of three polarization curves.

**Figure 10 materials-16-04323-f010:**
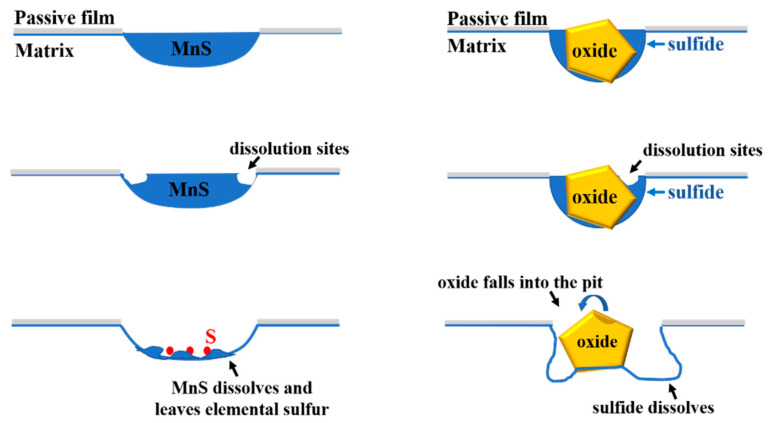
A brief schematic of dissolution process of MnS and oxy-sulfide inclusion.

**Table 2 materials-16-04323-t002:** Chemical composition of re-sulfurized 304 stainless steel (wt. %).

Element	C	Si	Mn	P	S	Ni	Cr	Ti	Al	Fe
Re-sulfurized SS304	0.073	0.72	2.81	0.022	0.043	8.17	18.1	0.011	0.013	balance

**Table 3 materials-16-04323-t003:** *E_corr_* and *i_corr_* values for matrix, matrix containing MnS, matrix containing oxy-sulfide.

	Matrix	MnS	Oxy–Sulfide
*E_corr_*	−302.7 mV	−311.2 mV	−290.7 mV
*i_corr_*	5.689 pA	6.731 pA	4.282 pA

## Data Availability

Restrictions apply to the availability of these data. Data were obtained from Dan Li and are available with the permission of Dan Li.

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
