# Peer review of "Localized Corrosion of Stainless Steel Triggered by Typical Inclusions in NaCl Solution: Oxy-Sulfide and MnS"

_materials, 2023, doi:10.3390/ma16124323_

Round 1
Reviewer 1 Report
In this paper, localized corrosion behavior of stainless steel (SS) induced by
typical inclusion[s] in NaCl solution was assessed. immersion tests and microelectrochemical tests were employed for this purposes. It is shown inclusions increase susceptibility of local area to pitting corrosion. it is also shown that Oxy-sulfide exhibits a complex electrochemical behavior in the passive region, due to its complex composition and multi-interface coupling effects. The paper could be considered for publications the journal of Materials after the following mandatory revisions:
1-The language of the paper in some parts requires revision.
2-The title of the paper is a bit strange. Perhaps using “inclusions” is unnecessary. Title needs a rewrite.
3-Abstract needs to be modified. Start the abstract with what has been done and then talk about the parameters studied and the main results. Use only one “inclusions” word in a sentence.
4-Introduciton should be strengthened and a bit extended. Some old references are used. To modify this section the following documents can be consulted:
-(2023). Ambient-stable polyethyleneimine functionalized Ti3C2Tx nanohybrid corrosion inhibitor for copper in alkaline electrolyte. Materials Letters, 337, 133979. doi: https://doi.org/10.1016/j.matlet.2023.133979
-(2023). Benzothiazole derivatives-based supramolecular assemblies as efficient corrosion inhibitors for copper in artificial seawater: Formation, interfacial release and protective mechanisms. Corrosion Science, 212, 110957. doi: https://doi.org/10.1016/j.corsci.2022.110957
5-“2. Experimental Procedure”.Also, “Materials” should be added.
6-is it wt% or at% in table 2?
7-figure captions should be reflective of each figure. Like what is figure 1a, b …. Captions should be modified for all figures
8-Consult the following documents in the discussion section.
- First-Principles Study on the Adsorption Characteristics of Corrosive Species on Passive Film TiO2 in a NaCl Solution Containing H2S and CO2. Metals. 2022; 12(7):1160. https://doi.org/10.3390/met12071160
9-what is the difference between figure 4a,b and fc,d?
10-conclusions should be bullet points.
11-better describe the rationale of the work at the end of the introduction. The last paragraph of introduction should better describe the reason for undertaking this research.
1-The language of the paper in some parts requires revision.
2-The title of the paper is a bit strange. Perhaps using “inclusions” is unnecessary. Title needs a rewrite.
Author Response
We thank the reviewers for their professional suggestions on our manuscript and responses to your comments have been listed one-by-one in an additional document.

Reviewer 2 Report
This paper looks at the influence of intermetallics on the corrosion of stainless in NaCl at neutral pH and so is very relevant and important.
There are some typos in the document so needs re reading to remove them. I have highlighted some on the attached doc with sticky notes.
The figure legends need more detail so that the reader can understand what they are showing without looking at the main text.
The authors refer to the MnS as rod shaped. Are they more likely to be discs?

English is good. Some typos which I have mentioned above.
Author Response
We thank the reviewers for their professional advice on our manuscript and the responses to your comments have been itemized in a separate document.

Reviewer 3 Report
One of the most severe forms of local corrosion is pitting corrosion, which mainly occurs when stainless steels come into contact with chlorine-containing solutions. The topic is well-known and widely researched. But they will continue because there is no complete solution to the problem of pitting corrosion in stainless steels.
Too many literary sources are cited when the work is not comprehensive. Maybe some should drop out. It is not entirely clear what the purpose of the present study is.
The description of the methods used needs to be refined. The cyclic potentiodynamic method is mainly used in the study of pitting corrosion. Why did the authors use only the classical potentiodynamic method? Why was this potential scan rate chosen? In such studies, it is advisable to follow the behavior of the material at different scan scan rates.
The texts below the figures need to be censored. The text under fig. 6 is given as SEM micrographs, but contains those from other studies, such as AFM for example, which is not acceptable.
To explain in more detail why electrochemical measurements are called micro electrochemical measurements.
The conclusion should be revised. The results obtained should be clearly described and the contributions of this research should be given.
English is understandable. Minor spelling corrections needed.
Author Response

(The authors gave the same response as above.)

Round 2
Reviewer 1 Report
The paper can be published in its revised format.